# K–8 Classroom Self-Collection Using XpressCollect Nasal Swab: A Usability and Efficacy Study

**DOI:** 10.3390/diagnostics12051245

**Published:** 2022-05-17

**Authors:** Kate Cox, Laura Haggerty, Kate Abeln, Luis Hernandez, Jessica Wicker, Leah Padgett, Mary Beth Privitera

**Affiliations:** 1HS Design, Morristown, NJ 07960, USA; laura@hs-design.com (L.H.); k.abeln@hs-design.com (K.A.); mb@hs-design.com (M.B.P.); 2Quantigen, Fishers, IN 46038, USA; luis.hernandez@quantigen.com (L.H.); jessica.wicker@quantigen.com (J.W.); leah.padgett@quantigen.com (L.P.)

**Keywords:** nasal swab, self-collection, school testing, COVID-19, SARS-CoV-2 testing

## Abstract

This study demonstrates that students in kindergarten through eighth grade can use the XpressCollect nasal swab to self-collect a specimen under the guidance of a teacher. This phased study was conducted with parents, teachers, and students. Phases 1 and 2 were conducted as interviews with teachers and parents to assess the suitability of the XpressCollect for children in kindergarten through eighth grade. Additionally, teacher and parent feedback was obtained to develop and optimize the instructional materials for subsequent phases. In Phases 3 and 4, teachers guided small groups and full classes of students through the sample collection process with XpressCollect. The samples collected by the students were sent to a laboratory to analyze the effectiveness of specimen self-collection based on the presence of ribonuclease P (RNase P) on each nasal swab. The presence of RNase P enables disease determination; thus, student samples were analyzed for adequate or inadequate sampling. All students in kindergarten through eighth grade are capable of self-collecting an anterior nares specimen with XpressCollect, as the laboratory results identified acceptable RNase P Ct values for the samples collected in a classroom setting.

## 1. Introduction

Schools and school-based programs are fundamental for the healthy development and well-being of children and adolescents [1]. Therefore, school closures due to COVID-19 and the suspension of in-person learning have had a profound impact on students, families, and educators. Guidance from the American Academy of Pediatrics emphasizes the criticality of maintaining safe in-person learning [1]. The CDC has identified regular COVID-19 testing as an important safety measure to keep schools open for in-person learning and activities [2]. Implementing testing programs in schools protects staff and students who may otherwise not have access to testing. A pilot testing program conducted in 2020 found that weekly screening testing of students, teachers, and staff can reduce in-school infections by an estimated 50 percent [3]. This paper is intended to address challenges in school-based testing, specifically in the administration of self-collection by teachers for children in kindergarten through eighth grade.

Self-collection reduces the testing burden for healthcare personnel, preserves personal protective equipment, and limits exposure to others [4,5,6,7]. Recent studies have demonstrated that adult self-collection and parent sample collection with a nasal swab produces adequate samples for COVID-19 testing that are comparable to clinician collected nasal swab specimens [7,8,9]. Moreover, a recent study conducted by Waggoner et al. found that with an instructional video and handout, children ages 4–14 could also self-collect an adequate sample for SARS-CoV-2 detection [10]. However, students’ ability to self-collect using nasal swabs in a classroom setting with teacher guidance and the required support materials has not been well established. Research is needed to determine whether this sample collection method can be effectively implemented in a school setting to produce accurate diagnostic results using nasal swabs for the detection of COVID-19 or other respiratory illnesses.

The purpose of this study was to evaluate the XpressCollect nasal swab device for classroom use and determine what is required for the development of an XpressCollect Classroom System. This investigation included the development of all training and instructional materials necessary to support teacher-guided specimen self-collection for students in kindergarten through eighth grade. To accomplish this, a series of usability tests were conducted, including laboratory sample testing, with great success.

## 2. Materials and Methods

This research was completed in 4 phases in order to optimize the instructional materials before using them to support specimen self-collection from groups of students in grades K–8. Table 1 outlines the objectives and structure of each study phase. To assure that teachers had adequate support materials, an initial review of existing instructional materials was conducted with teachers using semi-structured interview techniques. This approach was also utilized for Phase 2 with revised instructional materials and the inclusion of parent interviews. All semi-structured interviews were conducted using an interview guide containing the topics, themes, and areas to be discussed. Questions addressed product use, observations, likes/dislikes, and key learning areas. This approach included flexibility in the order and manner in which questions were asked. During the interview, participants were encouraged to share their perspectives, experiences, and feedback as they interacted with a sample collection device and instructional materials.

The methods utilized in each phase of usability evaluations follow best practices and the international standard for the application of usability engineering for medical devices, IEC62366-1:2015 [11]. IEC62366-1 recommends formative studies prior to conducting a summative usability evaluation. In this research, Phases 1–2 are considered formative studies, and Phases 3–4 are considered summative usability studies. Furthermore IEC62366-1 requires 15 participants from each user group to evaluate the medical device for during a summative usability assessment. There are two user groups in the summative study (Phases 3–4), including a student user group with a breakdown from each grade level, as well as a teacher user group. Following the standard, the target class size of 20 students was appropriate and represented simulated use conditions for a small group and full classroom setting. Furthermore, a use-related risk analysis (URRA) was generated to inform critical use tasks that would or could result in serious harm if performed incorrectly or not at all. The URRA is compliant with ISO14971 [12] and IEC62366-1. IEC62366-1 requires manufacturers to include all potential use-related hazards and hazardous situations in a risk management process that complies with ISO14971.

With IRB approval, Phases 3 and 4 were conducted with a combination of students and teachers. Teachers were observed as they used the product and instructional materials to facilitate students’ self-collection. Students were observed using the XpressCollect system. Interactions between teachers and students were noted as they completed the steps of use. Semi-structured interviews with teachers followed using an interview guide containing the topics, themes, and areas to be discussed. During the interview, participants were encouraged to share their experiences using the system and feedback on the collection process. Phase 3 was designed to assess self-collection in small groups with a teacher-to-student ratio of 1:5. Phase 4 was designed to assess full classroom self-collection as guided by a teacher with a ratio of 1:20.

Table 2 describes the participants and their roles by phase. A total of 260 participants were enrolled in this study. All participants were recruited by Fieldwork, and the study was conducted in Fort Lee, NJ, for Phases 1–2, New York, NY, for Phase 3, and Chicago, IL, for Phase 4 in a simulated classroom environment.

### 2.1. Teacher Introduction (Phases 3 and 4)

During the teacher introduction sessions in Phases 3 and 4, the XpressCollect device and informational materials were presented to each of the teachers to enable familiarization with the system. Following simulated actual use, each teacher was able to choose which support materials they would use to educate and guide their respective grade of students. The test administration from HS Design did not make any suggestions on how the teachers should conduct the sessions or which instructional materials to use. This was done to replicate the actual use context wherein teachers have academic freedom to instruct their classroom in a manner they see fit.

### 2.2. Sample Collection (Phases 3 and 4)

During the sample collection sessions, the teacher participants led the group of students through the sample collection process with XpressCollect (SteriPack, Lakeland, Florida, USA, #60600). The test administrator from HS Design supported the distribution, collection, and tracking of student samples. This was completed in this manner to assure identification of samples and students were correctly matched. This requirement was to assure accurate results when correlating the students’ ability to self-collect and root cause of any use-related issue.

#### 2.2.1. Data Collection

The HFE (Human Factors Engineering) data collectors used a new data collection sheet for each session to capture observations, participant comments, and feedback on instructional materials. Video recordings were captured with the permission of the participants and parent/guardians (if applicable) to allow for further review and analysis. The recorded information was used to update, complete, and clarify the data collected as part of the post-study analysis. With the participants’ permission, video was streamed to colleagues who were not able to observe the study in-person and to parents in a viewing room.

Test data were not directly linked to individual participant identification. Rather, the data collected corresponded to assigned participant numbers. Participant numbers included a code corresponding to the user group or grade. All collected data (data collection sheets, video recordings, photographs, informed consent forms, and confidentiality agreements) are stored on HS Design’s secure server. Paper copies of forms were scanned into digital copies prior to shredding. All raw data will be maintained for a minimum of 2 years in a folder on the server within the project file that will only be accessible by the research team. All participant information was deidentified prior to disclosure in this paper.

#### 2.2.2. Data Analysis

Each use problem (use error, close call, and difficulty) that was observed by the HFE data collectors was noted and analyzed to determine the potential root cause of the problem. This was done by reviewing the video recordings to note any behaviors, comments from the student and/or teacher, or any other external influences that may have impacted the students’ performance during the session.

To analyze the effectiveness of specimen self-collection and analyze the impact of each observed use problem, each student participant’s swab was collected and sent to Quantigen Biosciences, a CLIA-certified and CAP-accredited laboratory, to be analyzed for the presence of ribonuclease P (RNase P). Detection of RNase P is used as a sample adequacy control to ensure that a sample is suitable and meets the requirements for testing. Student samples were analyzed for RNase P, and adequate sample collection was determined. For any sample deemed inadequate, video recordings of the associated participant were reviewed to identify potential root causes.

Additionally, subjective feedback from teachers was compiled and analyzed for key themes, and students’ anecdotal comments were organized into related groups to reveal use-related subjective feedback.

In addition to analyzing classroom usability, the HFE personnel consolidated and analyzed the test data as follows:Count the total number of each type of use error, close call, and difficulty associated with each task;Identify and describe the conditions and causes that led to use errors, close calls, and difficulties associated with tasks (i.e., perform a root cause analysis);Organize the anecdotal comments into related groups to reveal opinion patterns;Conduct root cause analysis on inadequate samples to determine factors that may have hindered successful sample collection.

#### 2.2.3. Laboratory Testing Methods

To undergo laboratory testing, samples are required to have two forms of patient identification on the collection device. The current XpressCollect utilizes a 1D barcode with human readable number and a QR code to associate the sample to a patient. Classrooms do not have access to barcode scanners; therefore, the problem of associating samples with students remains. For this phase, each swab was assigned a unique ID, which was then connected to the barcode number printed on the tube. These barcode numbers were then associated with the student participant ID. Without this labeling, the research team would not have been able to trace the samples back to the participants.

The collected samples were sent to Quantigen for detection of RNase P. This was achieved using a Biosearch Technologies UltraDx SARS-CoV-2 assay and a Thermo Fisher TaqMan RNase P assay [13]. The Biosearch Technologies assay tests for both SARS-CoV-2 and RNase P. The assay was created for ultra-high-throughput testing via the Nexar platform. The Nexar platform is currently in use at Northwell Health, and the EUA of this process is currently under review at the FDA. At the time of testing, Quantigen did not have the Nexar system but ran the same assay using a qPCR machine. Each dry swab sample was eluted into 400 µL of direct lysis buffer (2.5 mM TCEP, 2 mM EGTA), vortexed for 30 s, and heated at 90 °C for 10 min. The samples were passively cooled and tested using an UltraDx SARS-CoV-2 assay on a QuantStudio^TM^ 7 Flex Real-Time PCR system, with 384-well plate format. The UltraDx SARS-CoV-2 assay tests 3.8 µL of sample (0.95% of the total sample) and has an RT step, so it detects both RNA and DNA for the RNase P target. Data were analyzed using QuantStudio^TM^ Real-time PCR software with auto threshold and auto baseline parameters.

In addition, each sample was tested using a Thermo Fisher TaqMan RNase P assay, as previously described, except that we did not purify the nucleic acid but instead used 10 µL of the crude sample lysate (2.5% of the total sample) [13,14]. This assay does not include an RT step and thus only looks at DNA for the RNase P target.

A lower the CT value equates to a higher amount of RNase P detected. The acceptance criteria for adequate specimen collection for an individual sample was an RNase P Ct value < 40, with a population mean RNase P Ct value ≤ 32 [13].

### 2.3. Materials for Evaluation

Table 3 outlines each of the materials evaluated throughout each phase of this study. The instructional materials were revised, and various iterations were presented across phases. Please refer to the Appendix A (five-step and eight-step instructions; teacher guide) for additional detail. The XpressCollect device (Figure 1) was evaluated in all phases. Additionally, new concepts were presented for evaluation in Phases 2, 3, and 4 based on observations and feedback on XpressCollect. This included four conceptual prototypes (Figure 2) in Phase 2 and a modified swab (Figure 3 and Figure 4), referred to as concept B, in Phases 3 and 4.

## 3. Results

### 3.1. Phases 1 and 2

During Phase 1, teachers provided feedback on the usability of XpressCollect for students. It was concluded that future revisions of the instructions should emphasize the key actions required and not include animal characters, which could be distracting. Furthermore, a distinct colored box around each step could help teachers reference the instructions when guiding their students. Key product design recommendations included increasing the outer diameter for grip and utilizing color to differentiate the swab and tube.

Informed by the Phase 1 findings, two variations of the instructions and four new swab design concepts (Figure 2) were developed for teachers and parents to evaluate in Phase 2. Phase 2 product design insights included size recommendations for a swab and tube with a longer swab grip than the XpressCollect system. Parent and teacher feedback identified improvements to the instructions, which included adding keywords and illustrating swab collection from the right and left nostrils. There was a correlation between preferences for longer or shorter instructions and the grade level of the intended students, with eight steps suggested for younger students who may need more guidance. Parents and teachers were shown an animated instruction clip and an instructional video developed by researchers at Emory University for their study on children’s self-collection [10]. An instructional video featuring children using XpressCollect was identified as more relatable and effective as an instructional tool than an animation. This format was developed for classroom use in future phases.

### 3.2. Phase 3

During Phase 3, all of the students in kindergarten through eighth grade were able to collect a nasal specimen with an acceptable RNase P Ct value, with mean RNase P Ct values of 26.3 and 26.6 in the Biosearch Technologies and Thermo Fisher assays, respectively.

The observed use problems during the sample collection tasks did not have an effect on the students’ ability to collect a viable specimen for laboratory testing. However, the current XpressCollect packaging and closure design prevented 14 of the 36 students from being able to open the tube independently. Figure 5 shows a graph of the number of observed use problems that occurred for each task with XpressCollect. These data are further broken down by showing the number of problems each grade range experienced. The younger students experienced more use problems with opening the package and swab. 

Each teacher led the students together (3–5 students), step by step. This method is an effective approach for small groups of students.

### 3.3. Phase 4

During Phase 4, 174 out of 176 students in kindergarten through eighth grade were able to collect a nasal specimen with an acceptable RNase P Ct value across both assays and with both swab concepts. Figure 6 below summarizes the results of all samples collected and analyzed with both assays. Group A used the XpressCollect swab samples, and Group B used the modified swab samples.

Of the two students that did not achieve an acceptable RNase P Ct value, one student (1–8) did not receive an acceptable RNase P Ct value (Ct < 40) on one of their swabs with the Biosearch Technologies assay. This was potentially due to a technical error on the Biosearch Technologies assay. The other student (K–8) never successfully opened swab concept B to collect a sample from their nose, therefore receiving an ‘undetected’ RNase P Ct value.

The majority of observed use problems during the sample collection tasks did not have an effect on the students’ ability to collect a viable specimen for laboratory testing. However, the current XpressCollect closure design prevented many students from being able to open the tube independently. In total, 48 of the 176 participants were unable to open the tube independently. Of the 60 total participants in kindergarten through second grade, less than half were able to open the tube without assistance. These findings indicate that students in grades K–8 are capable of self-swabbing for lab testing; however students in grades K–2 will need assistance in opening XpressCollect. Figure 7 shows a graph of the number of observed use problems that occurred on each task. These data are further broken down by showing the number of problems each grade experienced. The younger students experienced more use problems with opening the package and swab.

During the sessions, each teacher led the students through the collection process step by step. All of the teachers for grades K through seventh grade had all of the students in the class perform each step together. The eighth grade teacher split the class in half and allowed the students to move at their own pace to perform the steps. During the post-session interview, teachers for kindergarten, first, and second grade stated that in their classrooms, they would facilitate sample collection in small groups, as opposed to the entire class at once.

Table 4 further summarizes the percentage of students that experienced a use problem removing the XpressCollect swab from the tube versus those who had no use problems. The data for students who experienced use problems are broken out further in the far-right column by showing the percentage of students that could not remove the swab from tube independently. This further illustrates that students in grades K–2 will require assistance.

### 3.4. Performance Comparison between XpressCollect and Modified Swab

A comparison of the number of use problems the students experienced in Phases 3 and 4 when using XpressCollect versus the modified swab concept B is shown in Figure 8 and Figure 9. The handle design of swab concept B improved the students’ ability to open the tube on their own. Only three students in Phase 4 could not open swab concept B independently, partly due to the prototype handle disconnecting from the swab cap.

### 3.5. Phase 3 and 4 Performance Comparison

To compare the influence of class size on the students’ performance, the observed use problems for each task from the students in Phases 3 and 4 are plotted in Figure 10. There were no significant differences in observed use problems when students were in a full classroom versus small groups.

## 4. Discussion

This study demonstrates that adequate samples can be collected under the guidance of a teacher for students in kindergarten through eighth grade. Student self-collection in the classroom with their teacher is an alternative to going to a designated area to be overseen individually. The classroom-based approach can be quicker and less resource-intensive. This would also be more feasible and less disruptive for schools with limited space. Additionally, young students may feel more comfortable with the process when guided by their teacher rather than a staff member with whom they are less familiar.

The iterative approach used in this study provided the opportunity to improve the instruction materials based on observations. The optimized materials were used for the full-classroom study in Phase 4. These materials included an instructional video, five-step instructions, eight-step instructions, and a teacher guide.

The instructional video showed four children in grades 4–6 completing the steps of collection with text descriptions. At the end of the video, the students offered their impressions of the process and how it felt. This video was informed by feedback from teachers in Phases 2 and 3 that students of varying ages would be most relatable, particularly for older students, who would not identify with younger actors. The teachers’ subjective feedback suggested making the process less intimidating for hesitant students.

Teachers were given the option of choosing between eight-step instructions and five-step instructions based on what would be most appropriate for their grade level. The eight-step instructions are most thorough, showing all of the steps. To be more concise, the five-step instructions do not show the initial steps of blowing nose, sanitizing hands, and opening the package. Of the teachers who participated in the study, only three elected to distribute the instructions to their students. Teachers who would use the instructions preferred the more comprehensive eight-step version. Some teachers suggested that they would use a poster or Smart Board to show the instructions to their students for convenience.

Teachers indicated that sample collection would be conducted in small groups or with the full class simultaneously. Facilitating collection in small groups is an effective approach for all grade ranges but may be more significant for the lower grade range (K–2), in which the students may require more guidance and assistance with opening the swab. After their sessions in Phase 4, teachers in grades K–2 indicated that they would facilitate collection in small groups. Teachers for grades 3–8 indicated that they would collect with all the students at one time. Although collection as a class would be most efficient, it may not be possible for smaller classrooms, where students are not spaced out enough to safely remove their masks and blow their noses. Regardless of group size, teachers would guide the students through the process step by step to ensure proper sample collection. Another approach identified was to have an area in the classroom designated as the collection area. This may be implemented, if necessary, based on increased student needs or the classroom layout.

Usability findings during the study with XpressCollect included difficulties opening the package, trouble twisting the cap off of the tube, and failure to fully close the tube after collection. These use issues were more frequent for students in grades K–5. Despite any usability problems, participants were able to self-collect. If implemented in grades K–2, students will need assistance opening the package and the tube because the dexterity and force required exceeded some students’ abilities in this age range. Overall, fewer usability challenges were observed with the modified swab (concept B). Most notably, the handle design of the modified swab made it much easier for the students to remove the swab from the tube. Common difficulties participants experienced with the modified swab included difficulty opening the exterior packaging, attempting to pull the swab out of the tube rather than unscrew, and failure to fully close the tube after collection. Participants were observed performing unintended behaviors when using the XpressCollect and modified swab that deviated from the instructions. These behaviors included touching the swab (before, during, or after swabbing their nose), moving the swab in a twisting motion inside their nose, and performing fewer than four circles with the swab inside their nose. 

In this study, the teachers guiding the student participants did not have a relationship before the session. As a result, they were not able to customize their approach for the children’s specific needs. In post-session interviews, teachers suggested that their students’ behavior may be different from the reserved demeanor of the study participants because their students would be in a familiar environment. Some teachers indicated that this would make the students more comfortable and sample collection easier, whereas others indicated that their students would display more behavioral issues during the process. Furthermore, teachers suggested that due to interacting with their students regularly, they are aware of which students may need more attention or assistance. Figure 11 illustrates the proposed workflow for using XpressCollect in a classroom. This workflow may be tailored to meet the needs of the specific class.

Integration with a web application or portal will enable schools to properly document and communicate test results. After the students self-collect and secure their tube, they will approach the teacher one at a time with their tube so that the teacher can associate that student’s collection tube with their name in the portal. After this pairing is complete, tubes can be collected. Teachers suggested that their school nurse or administrator would be responsible for sending the samples to the lab for testing.

### Study Limitations

The results of this evaluation only include the students’ first use of the XpressCollect swab and tube. As students become more familiar with the XpressCollect process, it is anticipated that use problems will decrease. For example, some students were initially unable to open the package because they did not know where or how to pull it apart. Once they learn how the package opens, they may not have issues in future uses. This is supported by recent research that studied K–8 self-collection over the course of a school year and found that with weekly testing, errors rapidly decreased within a month [15].

This study was intended to be qualitative in nature. As such, the small sample sizes utilized throughout each phase of the study do not allow for statistical analysis to be performed or produce any statistically significant data.

## 5. Conclusions

In conclusion, all students in kindergarten through eighth grade are capable of self-collecting an anterior nares specimen with XpressCollect to test for COVID-19. During the study, class size was found to have no bearing on the usability of the XpressCollect system. These conclusions are supported by laboratory results, which identified acceptable RNase P Ct values for the samples that the students collected in a classroom setting, suggesting that this method can produce accurate test results for COVID-19.

## Figures and Tables

**Figure 1 diagnostics-12-01245-f001:**
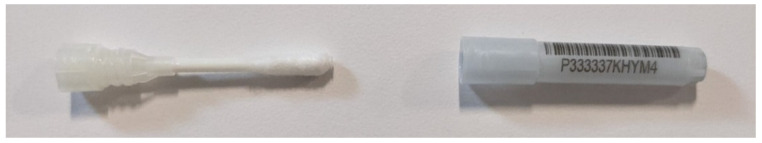
XpressCollect swab (**left**) and tube (**right**).

**Figure 2 diagnostics-12-01245-f002:**
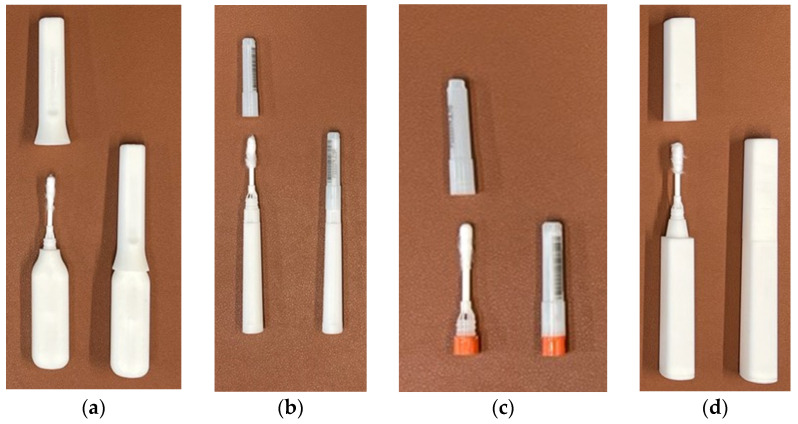
Prototype concepts: (**a**) key twist concept; (**b**) elongated concept; (**c**) colored cap concept; (**d**) marker concept.

**Figure 3 diagnostics-12-01245-f003:**
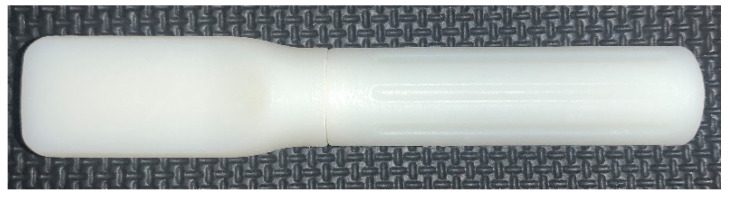
Concept B modified swab.

**Figure 4 diagnostics-12-01245-f004:**
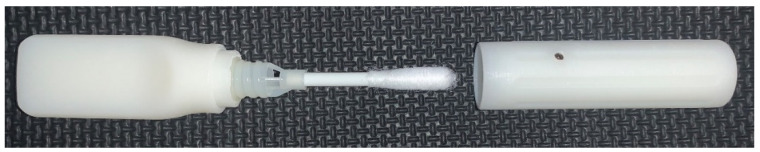
Concept B modified swab open.

**Figure 5 diagnostics-12-01245-f005:**
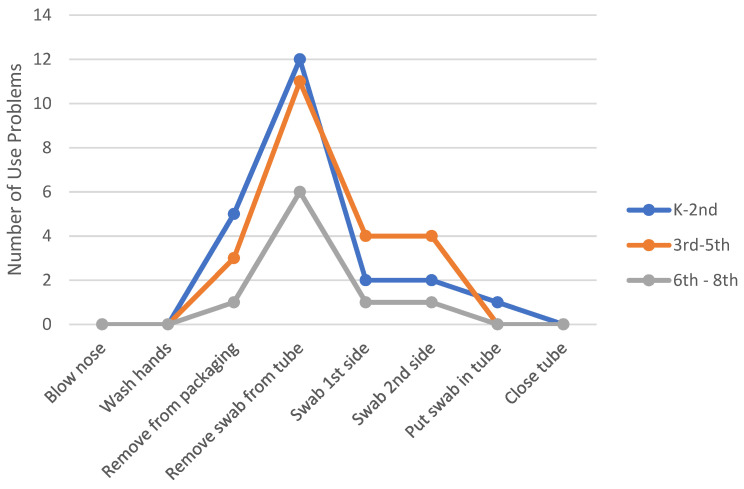
Number of use problems per task by grade range in Phase 3 using XpressCollect.

**Figure 6 diagnostics-12-01245-f006:**
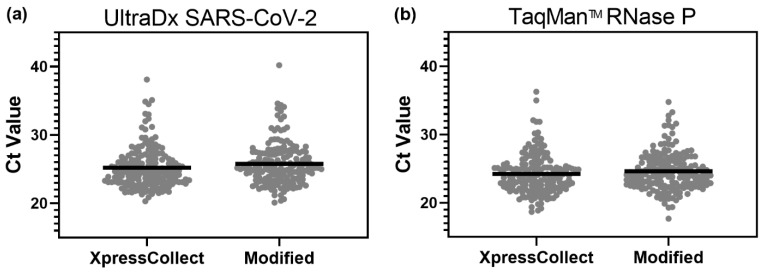
RNase P Ct values for 176 self-collected samples using the XpressCollect compared to the modified swab concept B, as detected by either the (**a**) UltraDx SARS-CoV-2 or (**b**) TaqMan^TM^ RNase P assay. The mean Ct value for each condition is marked by a horizontal line.

**Figure 7 diagnostics-12-01245-f007:**
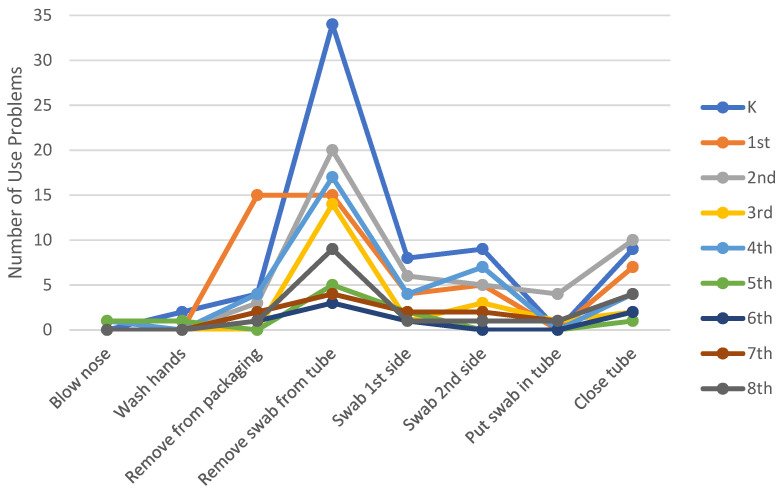
Number of use problems per task by grade in Phase 4 using XpressCollect.

**Figure 8 diagnostics-12-01245-f008:**
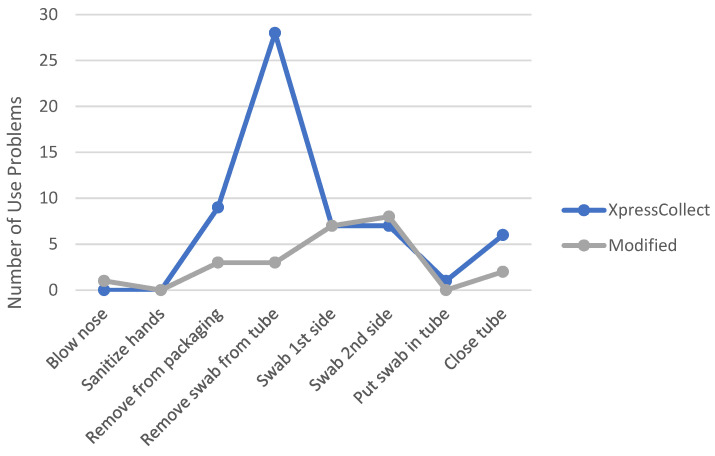
Comparison of observed use problems between XpressCollect and modified swab concept B in Phase 3.

**Figure 9 diagnostics-12-01245-f009:**
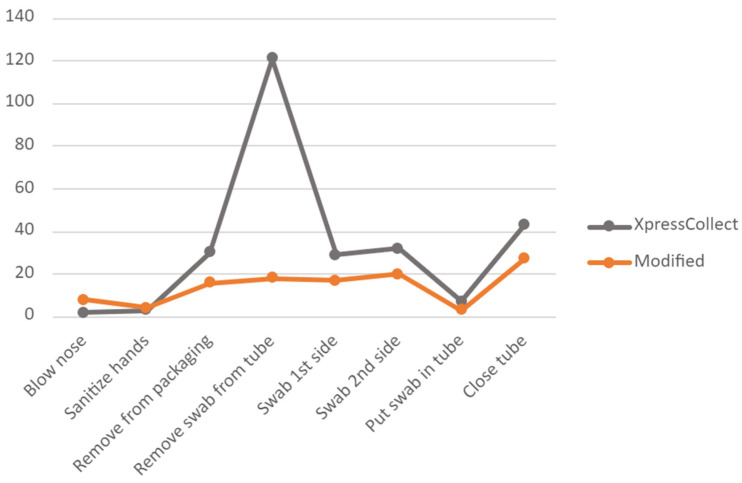
Comparison of observed use problems between XpressCollect and modified swab concept B in Phase 4.

**Figure 10 diagnostics-12-01245-f010:**
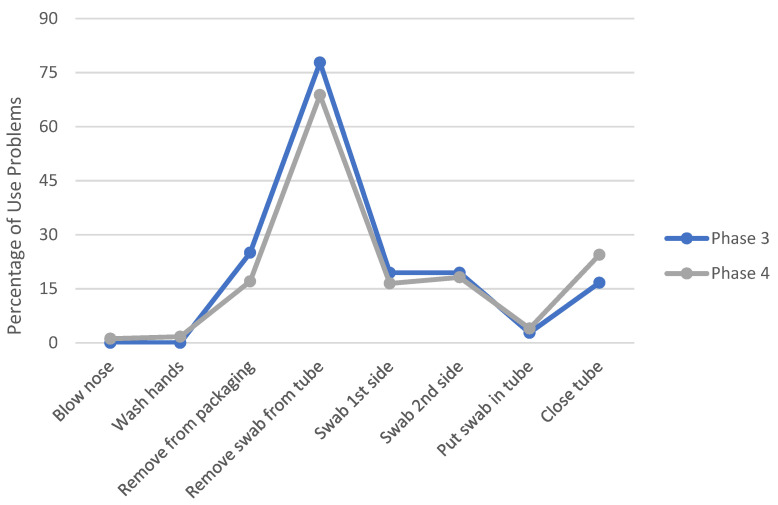
Percentage of students experiencing use problems per task across Phases 3 and 4 using XpressCollect.

**Figure 11 diagnostics-12-01245-f011:**
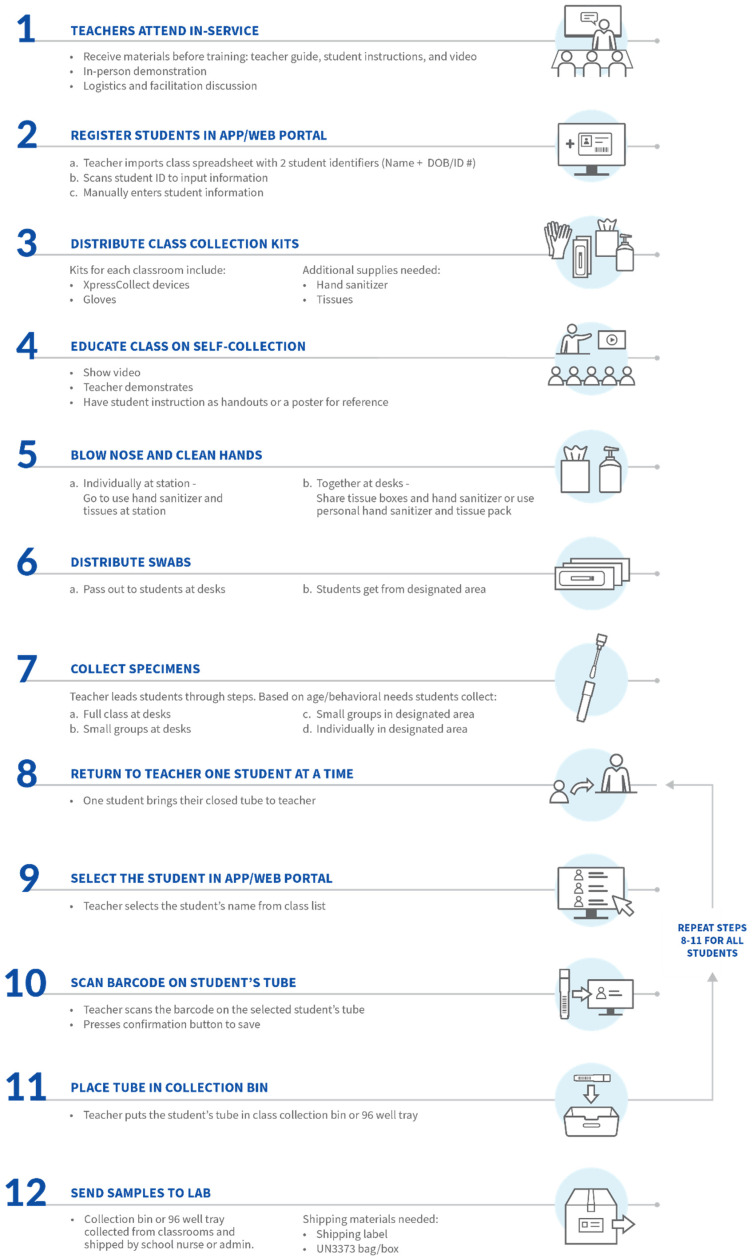
Proposed workflow for classroom self-collection with XpressCollect.

**Table 1 diagnostics-12-01245-t001:** Phased Study Objectives.

Study Phase	User Group	Objectives
Phase 1—Initial Review	Teacher	Determine what is needed to create clear instructions for students in grades K–8 (e.g., graphics, amount of text to include). Determine the process of distributing and collecting samples in the classroom. Determine whether modifications to the XpressCollect system are necessary in order to support classroom use. Determine the optimal way to communicate the results.
Phase 2—Secondary Review	Teacher	Determine what is needed to create clear instructions for students in grades K–8 (e.g., graphics, amount of text to include). Determine the process of distributing and collecting samples in the classroom. Determine whether improvements to instructions are necessary to be effective for the intended users (school children). Determine if modifications to the XpressCollect system are necessary in order to support classroom use. Assess the instructional materials for implementation within a classroom setting. Determine the optimal way to communicate the results.
	Parent	Determine what is needed to create clear instructions for children.
Phase 3—Initial Classroom Review	Teacher	Assess the instructional materials for implementation within a classroom setting.
	Student	Determine whether teachers are able to guide students in grades K–8 in self-collection using the XpressCollect system. Determine the process of distributing and collecting samples in the classroom. Determine whether students can successfully use the XpressCollect system.
Phase 4—Final Classroom Review	Teacher	Determine if the XpressCollect system is able to be used in a classroom setting. Determine whether the instruction materials are effective.
	Student	Determine whether students can successfully use the XpressCollect system.

**Table 2 diagnostics-12-01245-t002:** Sample size and distribution by phase.

Study Phase	User Group	Role	Sample Size	Distribution
Phase 1	Teacher	Participate in individual interviews	10	K–2: *n* = 3 3–5th: *n* = 5 6–8th: *n* = 2
Phase 2	Teacher	Participate in individual interviews	8	K–2: *n* = 3 3–5th: *n* = 2 6–8th: *n* = 3
	Parent	Participate in individual interviews	4	K–2: *n* = 2 3–5th: *n* = 2 6–8th: *n* = 0
Phase 3	Teacher	Attend teacher introduction; execute student testing in group sessions of 3–5 students	8	K–2: *n* = 3 3–5th: *n* = 2 6–8th: *n* = 3
	Student	Participate in group sessions of 3–5 students self-swabbing	36	K–2: *n* = 12 3–5th: *n* = 12 6–8th: *n* = 12
Phase 4	Teacher	Attend teacher introduction; execute student testing in group sessions of 20 students	18	2 participants per grade (K–8th)
	Student	Participate in group sessions of 20 students self-swabbing	176	K: *n* = 20 1st: *n* = 17 2nd: *n* = 20 3rd: *n* = 19 4th–8th: *n* = 20

**Table 3 diagnostics-12-01245-t003:** Materials evaluated during the associated study phases.

Phase	Materials Evaluated
Phase 1	XpressCollect
	Animal cartoon instructions
	Instructions depicting hands
	Instructions without words
Phase 2	XpressCollect
	Conceptual prototypes ^1^
	Five-step instructions
	Eight-step instructions
	Animation of swabbing process
	Instructional video with children
	Teacher guide
Phase 3	XpressCollect
	Modified swab (Concept B)
	Five-step instructions
	Eight-step instructions
	Instructional video with children
	Teacher guide
Phase 4	XpressCollect
	Modified swab (Concept B)
	Five-step instructions
	Eight-step instructions
	Instructional video with children
	Teacher guide

^1^ See Figure 2 for images of conceptual swab prototypes.

**Table 4 diagnostics-12-01245-t004:** Students that experienced use problems removing the XpressCollect swab from the tube.

Grade	Percentage of Students Who Had No Use Problems	Percentage of Students with Use Problems Removing Swab from Tube (Use Error or Difficulty)	Percentage of Students Who Could Not Remove Swab from Tube Independently
Kindergarten	10%	90%	75%
1st	12%	88%	71%
2nd	25%	75%	60%
3rd	42%	58%	5%
4th	15%	85%	10%
5th	60%	40%	0%
6th	80%	20%	0%
7th	85%	15%	15%
8th	55%	45%	10%

## Data Availability

The data presented in this study are available on request from the corresponding author.

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
