# Peer review of "K–8 Classroom Self-Collection Using XpressCollect Nasal Swab: A Usability and Efficacy Study"

_diagnostics, 2022, doi:10.3390/diagnostics12051245_

Round 1

Reviewer 1 Report

Although the authors tried to evaluate the usability and efficacy of self-collecting device of nasal swab among K-8, this paper lacks of scientific interest. The only positive finding looks the difficulty of opening the screw of the collection kit, which may be easily resolved by the explanation of the teachers. The photos of classroom do not seem necessary. 

Author Response

Response to Reviewer 1 Comments

Point 1: Although the authors tried to evaluate the usability and efficacy of self-collecting device of nasal swab among K-8, this paper lacks of scientific interest. The only positive finding looks the difficulty of opening the screw of the collection kit, which may be easily resolved by the explanation of the teachers. The photos of classroom do not seem necessary.

 Response 1: Classroom photos have been removed.

Reviewer 1 asserts that the manuscript lacks of scientific interest, however this study is intended to be qualitative in nature as it follows best practices and the international standard IEC62366-1 :2015 Medical devices — Part 1: Application of usability engineering to medical devices. 

Reviewer 2 Report

This study was performed to determine whether students in kindergarten through 8th grade could use XpressCollect nasal swabs to self-collect a specimen under the guidance of a teacher. The study was carried out in 4 phases.  The initial phases, 1 and 2, were conducted as interviews with teachers and parents to assess the suitability of the XpressCollect for children in kindergarten through 8th grade. Feedback was obtained to develop and optimize instructional materials to be used in the follow-on phases. These materials included an instructional video, 5-step and 8-step instructions, and a teacher guide. In phases 3 and 4, teachers guided small groups of students and full classes of students through the sample collection process. Samples were analyzed  for the presence of Ribonuclease P on each nasal swab. Adequate or inadequate sampling was determined based on an acceptable Ct value for each sample. The conclusion was that the students in all grades are capable of self-collecting an anterior nares sample.

Specific comments

A nice, well- executed study

Manuscript is well written

Author Response

Response to Reviewer 2 Comments

Point 1: This study was performed to determine whether students in kindergarten through 8th grade could use XpressCollect nasal swabs to self-collect a specimen under the guidance of a teacher. The study was carried out in 4 phases.  The initial phases, 1 and 2, were conducted as interviews with teachers and parents to assess the suitability of the XpressCollect for children in kindergarten through 8th grade. Feedback was obtained to develop and optimize instructional materials to be used in the follow-on phases. These materials included an instructional video, 5-step and 8-step instructions, and a teacher guide. In phases 3 and 4, teachers guided small groups of students and full classes of students through the sample collection process. Samples were analyzed  for the presence of Ribonuclease P on each nasal swab. Adequate or inadequate sampling was determined based on an acceptable Ct value for each sample. The conclusion was that the students in all grades are capable of self-collecting an anterior nares sample.

Specific comments

A nice, well- executed study

Manuscript is well written

Response 1: We request clarity on the paragraph provided in the comments/suggestions section. Is this intended to be a suggested revision to the Abstract?

Reviewer 3 Report

COMMENTS TO AUTHORS

Congratulation for the work performed while the efforts to review a self collection testing method for very young individuals and to evaluate the procedure followed aiming to to produce a homogenous informational conclusion for its usability.

However I have some comments in order to assist you to improve the significance of your work.

Comment 1:

The manuscript reviewed  describe actually 3 completely different projects, the preparation of the relevant instruction material, the usability and the performance evaluation of the system studied. I am afraid that none of the segments was objectively evaluated in order to safe conclusions to be drawn. There is no any statistical evaluation, while there is no any referred procedure to  assess the sensitivity of the method under investigation through repeated sampling from the same subject.

Comment 2:

The title does not correspond to the study while K-8 participants were less than 20  out of 176 in phase 4. Therefore must be rephrased.

Comment 3:

The instruction material even that it was meticulously prepared was never studied in comparison to a standardized guide leaflet or to a previous collection in  order to be proven effectively efficient enough. Nevertheless almost half of the students could not open the studied tube independently (lines 220-2202)

Comment 4:

The very small sample size in phases 1 through 3 cannot exclude biased decisions made in regard of the system usability and the instructor skills (Table 2).

Comment 5:

The subject cohort in phase 4 was small and heterogeneous (Table 2) for safe interpretation of the results (Figure 9 and Table 4).

Comment 6:

There are not referred any study limitations within the discussion section.

Comment 7:

Conclusion must be rephrased in accordance with  the results presented in the manuscript.

Author Response

Response to Reviewer 3 Comments

Point 1: The manuscript reviewed describe actually 3 completely different projects, the preparation of the relevant instruction material, the usability and the performance evaluation of the system studied. I am afraid that none of the segments was objectively evaluated in order to safe conclusions to be drawn. There is no any statistical evaluation, while there is no any referred procedure to  assess the sensitivity of the method under investigation through repeated sampling from the same subject.

Response 1: Reviewer 3 asserts that the manuscript describes 3 different projects, however each phase was designed and intended to build onto one another and produce a set of materials that could be used by K-8 students with varying reading levels.

Point 2: The title does not correspond to the study while K-8 participants were less than 20 out of 176 in phase 4. Therefore must be rephrased.

 Response 2: We request additional information and clarity as to how the title does not correspond to the study. Phase 4 was conducted with students in each grade, Kindergarten through 8th grade.

Point 3: The instruction material even that it was meticulously prepared was never studied in comparison to a standardized guide leaflet or to a previous collection in  order to be proven effectively efficient enough. Nevertheless almost half of the students could not open the studied tube independently (lines 220-2202).

Response 3: Reviewer 3 asserts that the instruction material was not proven effectively enough. However, the authors make no claim that the instruction materials developed are better than another form of instructions. The intent was to provide relevant information / materials to assist the participants. Options were provided (5 vs 8 step) to allow participants to choose the preferred embodiment offered. The intent of the study was to see if students could operate the device to self-collect a sample.

Point 4: The very small sample size in phases 1 through 3 cannot exclude biased decisions made in regard of the system usability and the instructor skills (Table 2).

Response 4: Agree – Added study limitations subsection into the discussion section of the manuscript.

Point 5: The subject cohort in phase 4 was small and heterogeneous (Table 2) for safe interpretation of the results (Figure 9 and Table 4).

Response 5: We request clarity about what was meant by the subject cohort being ‘heterogeneous’. We structured the study to be as close to a real classroom environment, with each child in the class being of the same school grade level. The sample size in each session also reflects a class size in the United States and is appropriate for usability testing per IEC62366-1 :2015 Medical devices — Part 1: Application of usability engineering to medical devices. 

Point 6: There are not referred any study limitations within the discussion section.

Response 6: Agreed – added a study limitations subsection within the Discussion section of the manuscript. This aligns with comment 4.

Point 7: Conclusion must be rephrased in accordance with the results presented in the manuscript.

Response 7: We request additional information and clarity about how the conclusion does not currently reflect the qualitative results presented in the manuscript.

Round 2

Reviewer 1 Report

Reviewers' comments are well revised and the manuscript is improved.

Author Response

Point 1: Reviewer’s comments are well revised and the manuscript is improved.

Response 1: Thank you for your time and feedback

Reviewer 2 Report

no issues

Author Response

Point 1: no issues

Response 1: Thank you for your time and feedback

Reviewer 3 Report

Your contribution with the description and evaluation of a novel simple procedure, which may be adopted as a valuable tool for the younger generation, is considered very relevant to the needs of the society in a pandemic status.

Author Response

Point 1: Your contribution with the description and evaluation of a novel simple procedure, which may be adopted as a valuable tool for the younger generation, is considered very relevant to the needs of the society in a pandemic status.

Reviewer 3 rated the research design and method description ‘must be improved’.

Response 1: Reviewer 3 asserts that the research design and method description must be improved. We have added detail to justify and describe the methods used in this study and appropriateness of the research design per international standards for usability and risk management for medical devices, IEC62366-1 and ISO14971, respectively.